DOI: 10.1038/s41467-017-00822-y　　**OPEN**

# Large three-dimensional photonic crystals based on monocrystalline liquid crystal blue phases

Chun-Wei Chen[1,2], Chien-Tsung Hou[1], Cheng-Chang Li[1], Hung-Chang Jau[1], Chun-Ta Wang[1], Ching-Lang Hong[1], Duan-Yi Guo[1], Cheng-Yu Wang[1,2], Sheng-Ping Chiang[1], Timothy J. Bunning[3], Iam-Choon Khoo[2] & Tsung-Hsien Lin[1]

Although there have been intense efforts to fabricate large three-dimensional photonic crystals in order to realize their full potential, the technologies developed so far are still beset with various material processing and cost issues. Conventional top-down fabrications are costly and time-consuming, whereas natural self-assembly and bottom-up fabrications often result in high defect density and limited dimensions. Here we report the fabrication of extraordinarily large monocrystalline photonic crystals by controlling the self-assembly processes which occur in unique phases of liquid crystals that exhibit three-dimensional photonic-crystalline properties called liquid-crystal blue phases. In particular, we have developed a gradient-temperature technique that enables three-dimensional photonic crystals to grow to lateral dimensions of ~1 cm (~30,000 of unit cells) and thickness of ~100 μm (~ 300 unit cells). These giant single crystals exhibit extraordinarily sharp photonic bandgaps with high reflectivity, long-range periodicity in all dimensions and well-defined lattice orientation.

[1] Department of Photonics, National Sun Yat-sen University, Kaohsiung 80424, Taiwan. [2] Department of Electrical Engineering, The Pennsylvania State University, University Park, PA 16802, USA. [3] Materials and Manufacturing Directorate, Air Force Research Laboratory, Wright-Patterson AFB, OH 45433, USA. Chun-Wei Chen and Chien-Tsung Hou contributed equally to this work. Correspondence and requests for materials should be addressed to I.-C.K. (email: ick1@psu.edu) or to T.-H.L. (email: jameslin@faculty.nsysu.edu.tw)

Three-dimensional (3D) photonic crystals, an optical analog of atomic lattices, are attractive materials for versatile manipulation of light[1–9]. In a 3D photonic crystal, the dielectric constant is tailor-made to vary periodically in three dimensions, giving rise to a so-called photonic bandgap which prohibits electromagnetic propagation and substantially modifies the dispersion around a specific wavelength (frequency) region. 3D photonic crystals and their variants that contain specifically designed defect structures to further modify their electromagnetic properties continue to attract intensive interest in the quest to realize efficient dispersion engineering, micro integrated circuits, mirrorless lasers, and other advanced photonic applications. One of the greatest challenges in realizing 3D photonic crystals for application in the optical domain at sub-micrometer wavelengths is the efficient fabrication of large-dimension (>1000's unit cells) crystals with high refractive index contrast. There have been many attempts to develop large periodic nanostructures including layer-by-layer photolithography, colloidal self-assembly, direct laser writing, and holographic lithography[2, 3, 6, 9, 10]. All processes and techniques employed so far are still laden with many technical and/or cost challenges, especially for cases where the photonic crystals are designed to work in the ultraviolet–visible spectrum.

Here we report the experimental realization of a truly 3D photonic crystal built from a unique phase of liquid crystal, namely blue-phase liquid crystal (BPLC)[11–34]. BPLCs are a special class of chiral nematics (also termed cholesterics) in which the director axes self-assemble into doubly twisted helices and exhibit three phases BPIII, BPII, and BPI in order of decreasing temperature from the isotropic liquid phase. BPIII is amorphous, whereas BPII and BPI are simple cubic and body-centered cubic (BCC), respectively, with lattice constants on the order of a few 100's nm. BPI and BPII thus exhibit selective Bragg reflections in the visible spectrum. Owing to their intrinsic liquid crystal properties, BPLCs are electro-optically active[14, 15, 20, 21, 24, 25] as well as highly nonlinear[16–19, 22, 23]. Although BPLCs as photonic crystals have been widely studied, most applications to date employ only polycrystalline or nearly amorphous structures[18, 20, 23, 24]. There have been some attempts to align BP polycrystals by surface treatment or externally applied electric fields[26–29]. Surface alignment, though proven effective in generating monodomain crystals, works only with the BPII phase in thin samples[27–29] (a few microns). Using an applied electric field[26], only the crystallographic axis parallel to the field is guaranteed, whereas the lattice orientations in other dimensions remain random, prohibiting application in many photonic systems where the lateral crystal orientation is also of crucial importance[5, 7, 8, 30] (e.g., preserving the transverse optical wavefront integrity of light). Additionally, randomly distributed lattices tend to degrade the quality factor and increase the scattering loss. According to Belyakov et al.[31], replacing the polycrystal with a single crystal enables the optical rotatory power to be enhanced by almost 40%. Others have shown that increasing the grain size diminishes the hysteresis during electro-optic switching and reduces the required driving voltage[32, 33]. For advanced photonic applications a sufficiently large monocrystalline photonic crystal is therefore highly desirable.

In this paper, we report the development of a gradient-temperature technique based on our detailed studies of the self-assembly and re-assembly processes in the ordered Blue Phases. This technique enables the fabrication of monocrystalline photonic crystals of unprecedented dimensions (lateral dimensions of ~1 cm, ~30,000's of unit cells, and thickness of ~100 μm, ~300's of unit cells) by controlling the natural self-assembly processes in BPLCs. Being able to increase the number of periods in a photonic crystal (N) not only extends the interaction length but also greatly improves the optical properties of a photonic crystal such as the photon density of states at the bandedge (proportional to $N^2$)[35]. A significant increase of $N$ from 100 to 10,000 will enhance the density of states by ~10,000 times, impacting group velocity, lasing threshold, spontaneous emission, and optical nonlinearity[4, 36, 37]. Furthermore, we have also demonstrated the possibility of polymer stabilization that enables not only a much wider operating temperature range of these crystals but also exceptional electrical tunability of their spectral properties. These giant single crystals exhibit extraordinarily sharp photonic bandgaps with high reflectivity, long-range periodicity in all dimensions and well-defined lattice orientation. The colossal, well-oriented BPLCs in either the polymer-free or polymer-stabilized form will serve as excellent templates for more advanced structures and diverse material systems[38–40], as well as topological platforms for molecular self-assembly and symmetry-protected topological photonic crystals[41, 42].

## Results

**Self-reassembly and merger of blue-phase platelets**. In the first part of the study focusing on a detailed investigation to gain insights into BPLC self-assembly and reassembly processes, a BP mixture **M1** (cf. "Methods") was employed as it exhibited visible reflections and sufficiently wide temperature intervals for both BPII and BPI formation. A 100 μm-thick cell containing **M1** placed in a temperature-controlled enclosure was first cooled from ISO phase at a rapid rate of 10 °C min$^{-1}$ to and held (quenched) at a specified temperature for 3 h before measurement (cf. "Methods"); the temperature/phase dependence of the average grain size obtained is shown in Fig. 1a. In the monophasic-BPII regime (abbreviated to mono-BPII; 37.7–36.5 °C), the platelets grew to ~80 μm (in lateral size) on average, with some of them as wide as 120 μm. The grain size dropped dramatically if the holding temperature departs from the mono-BPII phase to ~10 μm in mono-BPI (36.2–34.6 °C) phase. In order to gain more insights into the inner working of the growth process, the temporal evolutions of these BPII and BPI platelets, cf. Fig. 1b, c (Supplementary Movie 1) were studied. Upon rapid cooling from the ISO to BPII, the molecules self-assembled into small platelets of ~15 μm in diameter which then underwent multiple reassembly processes. The crystallographic axes of some platelets reoriented to accommodate their contiguous grains with elimination of the grain boundaries. After ~13 h of such repeated reassembly, the BPII platelets grew gradually to ~300 μm.

Similar procedures, however, barely yield improvement in the crystalline sizes grown from BPI platelets, similar to other studies employing surface alignment and slow cooling. The difficulty in growing large crystal from the BPI phase compared to BPII is attributable to the fact that within a lattice, all the line defects of BPII intersect with each other thus effectively acting as a single defect, whereas within the BPI phase, the line defects exist independently of one another[27, 43] (cf. the insets of Fig. 1b). The platelet-merger phenomenon follows the system's tendency to minimize free energy. When the lattice contains a sole defect (BPII), only one minimum of free energy exists, at which a single crystal is formed. In the case of BPI, there exist numerous defects in one lattice which correspond to many local free energy minima. This study implies that, a large single crystal can be grown only in the monophasic BPII temperature interval, cf. Fig. 1a. Such a platelet reassembly process involves the crystallographic-axis reorientation of an entire crystal; as the BPII platelets grows larger, the required energy for lattice rearrangement becomes larger until a local equilibrium occurs as indicated in Fig. 1b. We observed that several large BPII monocrystals of ~1 mm in lateral size can be obtained after the sample has sat for 1 week.

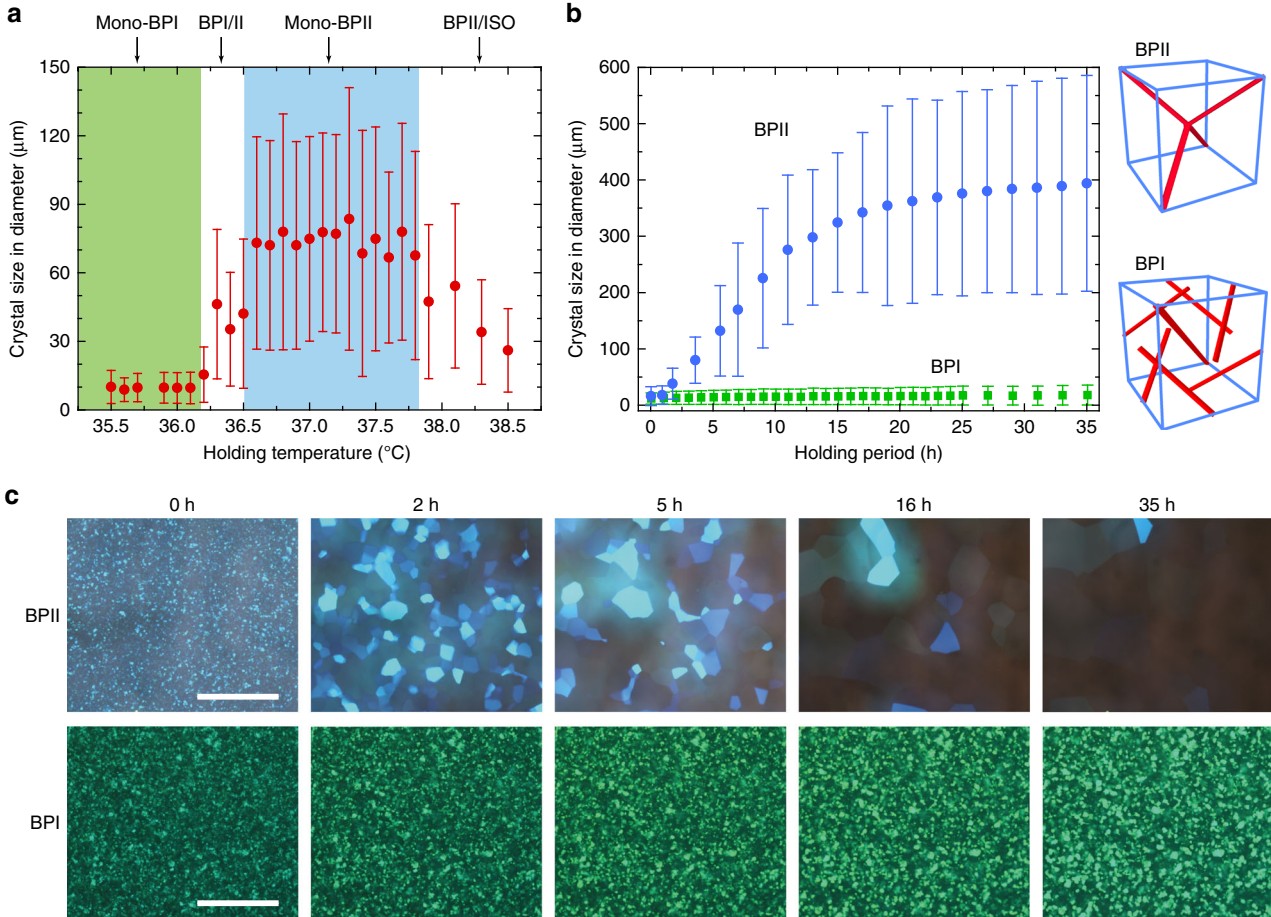

**Fig. 1** Self-reassembly of a BPLC. **a** Crystal size of BPLC **M1** as a function of holding temperature/phase, measured after holding the sample at the respective temperatures for three hours upon cooling from the ISO phase. **b** Sizes of BPII and BPI monocrystals as a function of holding time; the insets show the defect structures in unit cells of BPII and BPI, respectively. The data points for **a**, **b** were obtained by averaging the areal size of all platelets within the field of view; *error bar* indicates the deviation from the average size. Each platelet is approximated by an enclosing circle of diameter D. The deviation $\sigma$ is calculated by the formula: $\sigma = [(N-1)^{-1} \cdot \sum (D - D_0)^2]^{1/2}$, where $D_0$ is the average diameter and N is the number of platelets. **c** Microscope images of BPII and BPI crystals as a function of time (*scale bars*, 500 μm). See "Methods" for experimental details

Although large BPI single crystals cannot be prepared by direct self-reassembly as in the BPII case, we have discovered that using a well-formed BPII crystal as a precursor/mold it is possible to grow BPI single crystals of record-setting sizes. The capability to fabricate massive-sized BPI as well as BPII single crystals is important as it provides more choices of lattice structures (e.g., BCC structure and various non-cubic lattice structures of electrically distorted BPI[25]) and wider temperature ranges[11, 12]. Some important features of the growth process in the BPI phase are depicted in Fig. 2a–d (see "Methods" for more details on the successive quenching process). Figure 2b, c reveals that the contour of BPII platelets was completely preserved upon rapid annealing to BPI. Such pseudomorphic transformation commenced with the formation of striations in two directions generated from the mismatch between the two lattice structures[44] (Fig. 2c and Supplementary Fig. 1). These striped patterns also provide further information on the lattice orientation. By applying the Kössel diffraction technique to identify the lattice orientation, the bisector of the obtuse angle formed by the striations was identified as nearly matching the [200] axis of the BPI crystal (Supplementary Fig. 1). Large and uniform BPI single crystals were formed with high reflectivity and narrow bandwidth. Figure 2e, f also clearly indicates that over a wide field of view-area (~1 mm²), the reflection bandwidth of such an extreme-sized photonic monocrystal was found to be dramatically

narrower than that of a commonly used BP polycrystal—~11 nm as opposed to ~83 nm, approaching the theoretical expectation[45] (cf. Supplementary Note 2). Furthermore, a nearly mm²-sized BPI monocrystal (area $A \approx 0.9$ mm²; thickness $d \approx 100$ μm) shown in Supplementary Fig. 3 greatly exceeds the previously held record of $A \approx 0.2$ mm² with $d \approx 6$ μm[34].

**Crystal growth by gradient-temperature scanning**. To further push the size limit realized by these self-reassembled BP single crystals, a gradient-temperature scanning (GTS) technique was developed. Similar to Czochralski and float-zone processes[46], a spatial gradient of temperature was introduced in the conventional slow-cooling method. The GTS process utilizes the earlier-formed BPI platelets as seeds/nuclei for crystal growth. The system consists of a chamber enclosing a step motor and two temperature-controlled stages with a gap of ~2 mm in between (Fig. 3a). One of the heating stages is set at a temperature in the mono-BPII regime (designated as the high-$T$ stage), while the other is situated at a mono-BPI temperature (designated as the low-$T$ stage). Placing the cell across the two stages generates a gentle gradient of temperature. A micrometer step motor is used to move the sample from one stage to the other ensuring a controllable shift of the temperature gradient. Figure 3b displays an ~3 mm BPI monocrystal in a **M2** cell ($d \approx 100$ μm) grown by this technique with a scan rate of ~0.02 μm s⁻¹. A continuous

color variation from blue-green to green reveals the spatial temperature gradient from high to low temperature.

To achieve large crystal sizes in two dimensions, a "melt-and-regrow" strategy was devised (cf. Supplementary Note 4). After

growing to the desirable extent in one dimension (say $x$), the cell was rotated by 90°; one of the long edges of the grown BPI monocrystal was then melted (transitioned to the BPII phase) and allowed to grow in the other direction ($y$). In this manner,

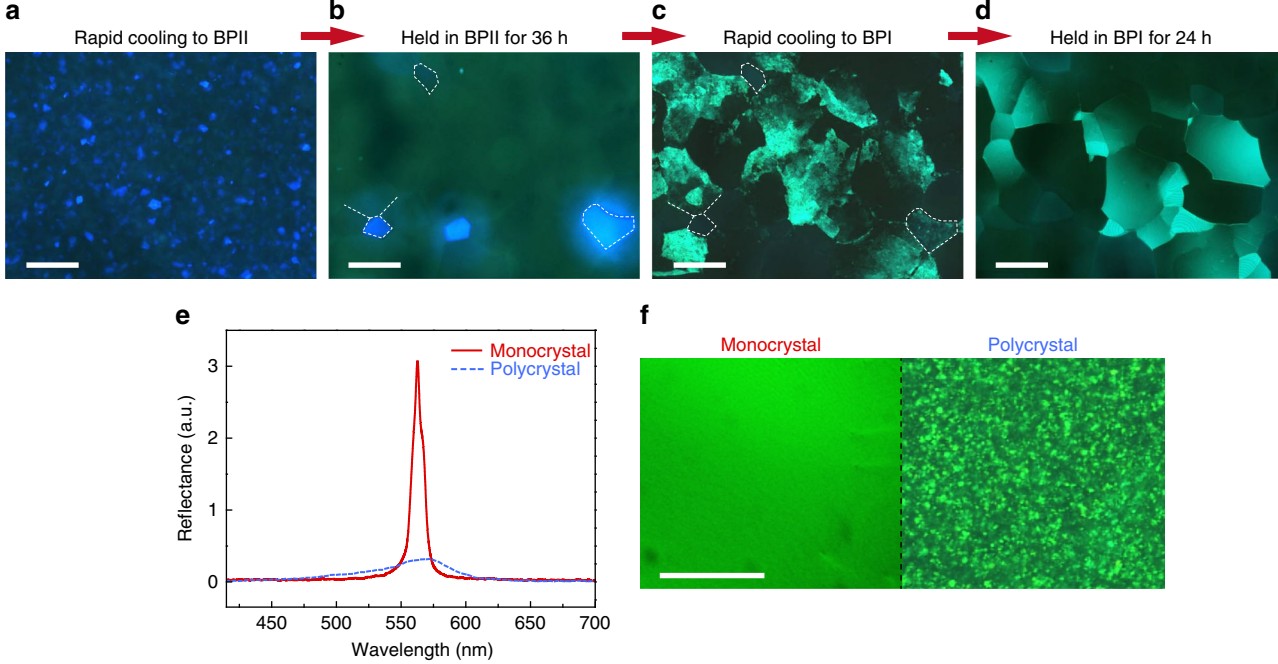

**Fig. 2** Formation of large BPI single crystals by self-reassembly and pseudomorphism. Formation process: microscopic images of **M1** (**a**), upon cooling to BPII from ISO (**b**), upon holding for 36 h in BPII (**c**), upon cooling to BPI (**d**), upon holding for 24 h in BPI (*scale bars*, 300 μm). Optical characterization of BPI monocrystal and polycrystal: **e** reflection spectra and **f** microscope images (*scale bar*, 100 μm). See "Methods" for experimental details

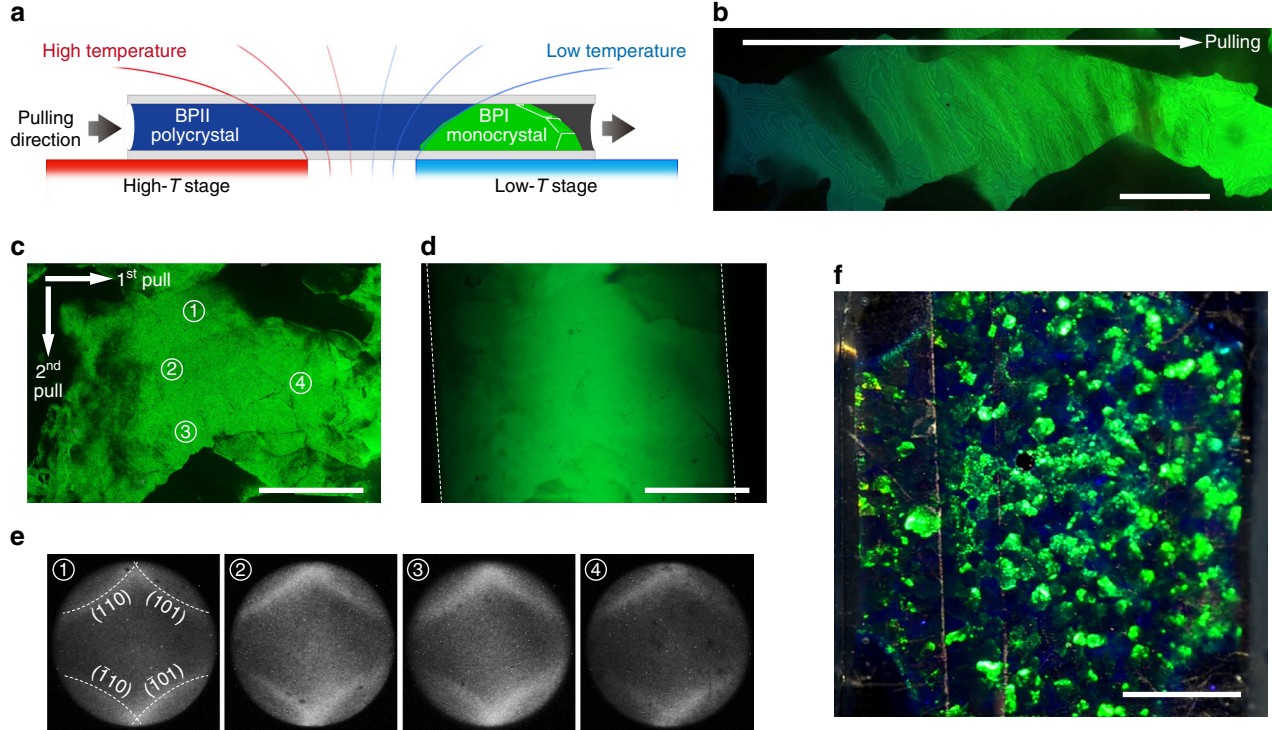

**Fig. 3** Formation of large BPI single crystals by gradient-temperature scanning. **a** Schematic depiction of the GTS system. **b** Microscope image of a 3 mm-long BPI single crystal fabricated by 1D GTS (reflection; *scale bar*, 500 μm). Microscope images of a 1.2 × 0.7 × 0.3 mm³-sized BPI monocrystal by 2D GTS in the **c** reflection and **d** transmission modes, respectively (*scale bars*, 500 μm). **e** Kössel diagrams captured at different labeled regions of the single crystal. **f** Macroscopic view of the sample with mm-sized BPI monocrystals (reflection; *scale bar*, 3 mm)

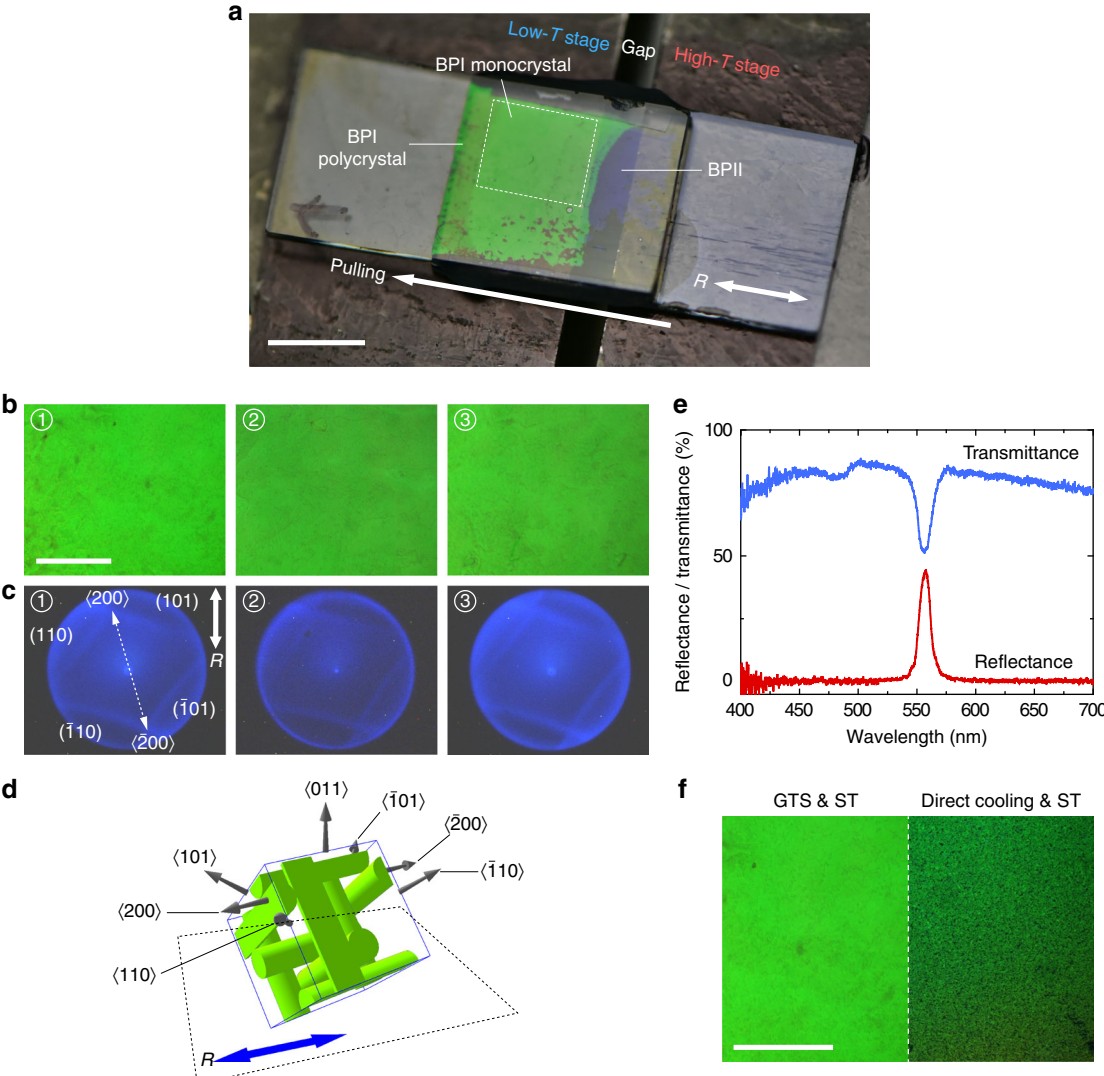

**Fig. 4** Surface-treatment-assisted gradient-temperature scanning growth. **a** Macroscopic view of a 100 µm-thick **M2** sample during GTS (*scale bar*, 5 mm). **b** Microscope images at different parts of the grown monocrystal (*scale bar*, 500 µm) and **c** respective Kössel diagrams. **d** Schematic depiction of the crystal orientation relative to alignment axis (*R*). **e** Transmission and reflection spectra of the single crystal. Note: the absence of a Darwin plateau at the peak reflection (or transmission dip) observed in their 1D counterpart (cholesteric liquid crystal) is simply due to the very small index difference in the BPLC unit cells compared to the index modulation in the cholesteric liquid crystal. **f** A set of microscope images comparing BPI crystals in the surface-treated (ST) sample grown by GTS (left; *scale bar*, 500 µm) and direct cooling (*right*)

a $1200 \times 700 \, \mu m^2$-area and 300 µm thick single BPI crystal was successfully grown, and it exhibits spectacular reflection and transmission properties as shown in Fig. 3c, d. The image taken in the transmission mode indicates a continuous lattice along the longitudinal axis of the cell. The Kössel diffraction technique was used to confirm the unity of lattice orientation within the platelet (Fig. 3e). Figure 3f shows a macroscopic view of a cell containing millimeter-sized BPI monocrystals, and Supplementary Movie 2 demonstrates the directional selective reflections from these "gigantic" 3D photonic crystals from different illumination angles.

To improve the uniformity of the grown BPI monocrystals, predetermination of lattice orientation is required and it is realized by treating the substrate with a rubbed polyimide layer. As illustrated in Fig. 3a, the temperature gradient exists not only in the scanning direction but also in the third (longitudinal) dimension, thereby influencing the nucleation of BPI to take place on the surface instead of the bulk. This enables the alignment layer to orient the nuclei effectively. Using this technique, a

perfectly oriented, nearly flat, $\sim 10 \times 5 \times 0.1 \, mm^3$-sized BPI monocrystal was grown. Figure 4a shows a crystal on the low-*T* stage (monodomain BPI), a variant across the temperature gap (BPI to BPII crystal), and a deep-blue crystal in the high-*T* stage region (BPII). Three regions of this large BPI crystal were randomly selected and microscope images (Fig. 4b) and Kössel diagrams (Fig. 4c) were captured. The number of terraces is significantly reduced and the Kössel diagrams are nearly identical, confirming the uniformity of the crystal (cf. Fig. 3 and Supplementary Fig. 3). The diffraction patterns also indicate that the grown crystal is arranged with its ⟨011⟩ direction parallel to the viewing axis and ⟨200⟩ direction placed at a small angle to the rubbing direction *R* (cf. Fig. 4d). Figure 4e plots the transmission and reflection spectra of one of the selected regions. The sharp transmission dip with a bandwidth of ~ 13 nm observed across a lateral extent of ~ 500 µm (and a longitudinal dimension of 100 µm that is equivalent to ~ 300 unit cells) indicates the uniformity of the monocrystal on the plane parallel to the substrate. It is noteworthy that the bandwidth of the

transmission dip is in good agreement with theoretical estimate (Supplementary Note 2).

The growth dynamics of a BPI crystal (Supplementary Movie 3) illustrates the possibility of "tuning" the photonic bandgap. The mixture employed is **M3**, the constituents of which are the same as **M2** but with higher chiral content. It is important

to note that even though the samples have been treated with surface alignment, without GTS, the BPI crystals grown by homogeneously annealing from BPII or ISO will be polycrystalline and randomly oriented, especially for thick cells (cf. Fig. 4f and ref. [27]). Additionally, with the aid of surface alignment, the GTS enables a higher growth rate of 0.1 μm s⁻¹ (compared to 0.02 μm s⁻¹ as mentioned previously).

To demonstrate that BPLCs as a 3D photonic crystal platform can adapt to different spectral regimes across the visible spectrum, three mm²-sized and ~mm-thick BPI monocrystals with red, green, and blue reflections were prepared by the surface-alignment-assisted GTS process (Fig. 5a, b). All three monocrystals are composed of the same constituents but having different mixing ratios (see "Methods" for more details). A blue-shift of the photonic bandgap is accomplished by increasing the concentration of the chiral agent. Figure 5b reveals that the photonic bandgap of each single crystal has a bandwidth of ~10–14 nm and reflectance of over 45% (Note: maximum reflection from the BPLC is 50% for linearly polarized probe light; the 5% loss originates primarily from specular reflection.).

**Polymer-stabilized blue-phase single crystal.** As first demonstrated by Kikuchi et al.[14], adding a polymer scaffold in situ through photopolymerization will thermodynamically stabilize a BPLC by templating the underlying lattice structure. Figure 6a demonstrates that these large monodomain BP crystals can also be so stabilized and do indeed exhibit a much wider temperature range (at least 25 °C) compared to their polymer-free counterparts (typically ~1–4 °C); temperature sensitivity of the photonic bandgap has also been suppressed, measuring $d\lambda/dT \approx 0.16$ nm °C⁻¹. The polymer-stabilized monocrystal possesses excellent dynamic tunability over a wide color range under direct current electric fields caused by field-induced distortion of the polymeric lattice scaffold[21]. Figure 6 reveals that the photonic bandgap can be linearly shifted with increasing field (~ 88 nm per V μm⁻¹ above the threshold at ~ 0.15 V μm⁻¹) from green to red. A dynamic tuning range of over 100 nm has been achieved with only ~1.3 V μm⁻¹. The lattice distortion is fully reversible, and the hysteresis between

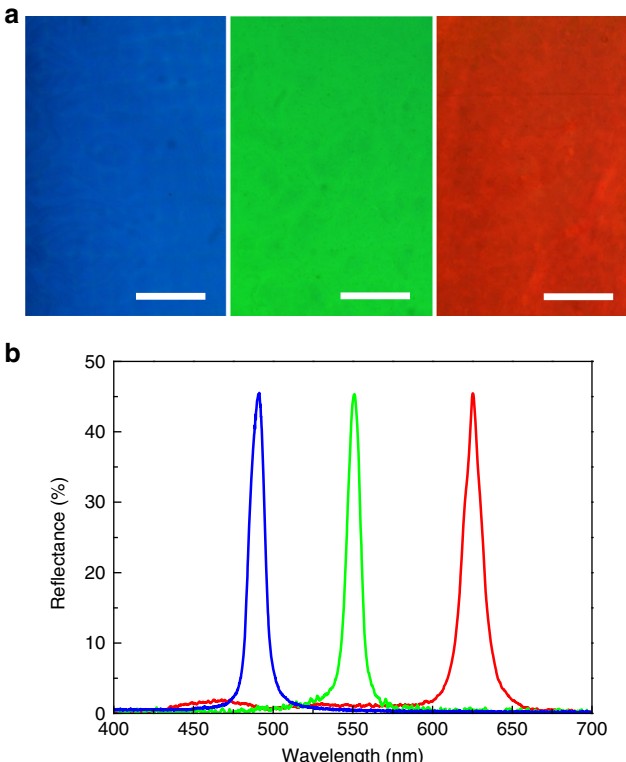

**Fig. 5** Blue, green, and red reflections from large BP single crystals. **a** Microscope images of three different BPI single crystals (*scale bars*, 200 μm) and **b** respective reflection spectra for linearly polarized probe light. See "Methods" for fabrication details

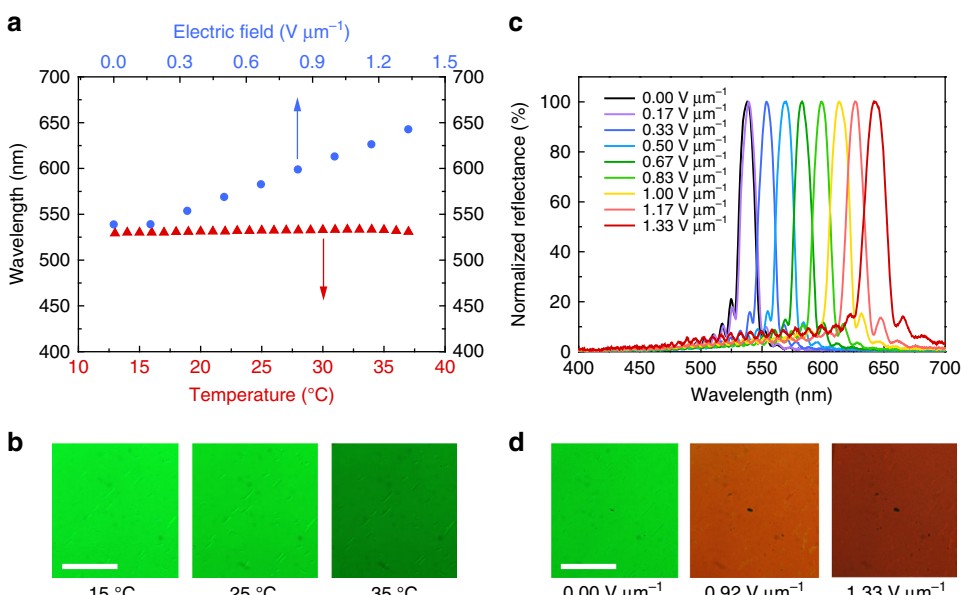

**Fig. 6** Temperature-invariant and electrically tunable photonic bandgap of large polymer-stabilized BP monocrystal. **a** Temperature and electric field dependences of the peak reflection wavelength. **b** Microscope images at different temperatures (*scale bar*, 500 μm). **c** Reflection spectra at different direct current field strengths. **d** Microscope images at different field strengths (*scale bar*, 500 μm)

increasing and decreasing field strength is almost negligible (cf. Supplementary Note 5). The success in polymer stabilizing a single BP crystal of such extraordinary size not only removes concerns regarding the narrow temperature range of BP but also clearly indicates the feasibility of BP-templated fabrication to yield flexible photonic-crystal gel systems[16, 20].

## Discussion

In summary, we have succeeded in developing monocrystalline 3D photonic crystals of centimeter length in lateral dimension and 100's of micrometers in thickness by a gradient-temperature growth methodology. Using BPII precursors, we demonstrate that small $\sim 10 \times 10\ \mu m^2$ single crystals can reassemble and merge into a single, much larger entity measuring almost $10^5$ times larger. Massive-sized monocrystalline BPI crystals can be grown using these BPII single crystals as a mold through pseudomorphosis and a gradient-temperature scanning technique. This shifts the mechanism of formation from kinetically controlled platelet merger to thermodynamically controlled heterogeneous nucleation. The resultant grain size is only limited by the length of the cell since no lattice reorientation and boundary removal is involved. Further introduction of surface alignment can guide the assembly of liquid crystal molecules during the crystal growth. By virtue of this surface-assisted GTS process, a 1 mm-wide BPI monocrystal can be grown within 3 h. Unprecedented gigantic 1 cm-wide single crystals (10,000's lattice spacings) of greater than 100 μm thickness are demonstrated. We have fabricated polymer-stabilized versions of these BP single crystals; the polymer-stabilized monocrystal possesses excellent dynamic tunability over a wide color range (~100 nm) under direct current electric fields, demonstrating the possibility of realizing versatile, reconfigurable, and highly thermodynamically stable photonic-crystal platforms. The ability to form an optical system with controllable on-demand optical properties based on a large-dimension 3D photonic crystal will serve as a catalyst for numerous applications.

## Methods

**Polymer-free BPLCs.** Three BPLC mixtures were employed in this study: **M1**, **M2**, and **M3**. **M1** is a mixture of 44.0 wt% nematic MDA-00-3461 (from Merck), 20.0 wt% nematic 5CB (from HCCH), and 36.0 wt% chiral smectic R811 (from Merck). On cooling from ISO, BPII nuclei appear at 38.5 °C, and the ISO coexists with BPII until the temperature decreases to 37.7 °C, below which the **M1** turns into monophasic BPII. BPI emerges at 36.5 °C, and the mono-BPI spans from 36.2 to 34.6 °C, below which the N* comes into sight. **M2** is composed of 32.0 wt% nematic HTW114200-050 (from HCCH), 33.0 wt% 5CB, and 35.0 wt% R811, showing a phase sequence of ISO-(35.6–34.5 °C)-BPII-(34.0–33.9 °C)-BPI-(29.8 °C)-N* on cooling. **M3** consists of 35.0 wt% HTW114200-050, 25.0 wt% 5CB, and 40.0 wt% R811, having a similar sequence to **M2** on cooling: ISO-(33.4–32.9 °C)-BPII-(32.7–32.6 °C)-BPI-(28.4 °C)-N*. **M4** consists of 30.0 wt% HTW114200-050, 37.0 wt% 5CB, and 33.0 wt% R811, exhibiting a phase sequence of ISO-(33.6–32.8 °C)-BPII-(32.1–32.0 °C)-BPI-(28.1 °C)-N*. Note that the measurements were done using 100 μm-thick BP samples, and the coexistence of BPI and BPII phases in the bulk was due to inhomogeneous heating of the temperature control system mK2000 (Instec).

**Polymer-stabilized BPLCs.** The precursor of the polymer-stabilized BP, designated as **M5**, is composed of 54.8 wt% HTW114200-050, 36.5 wt% R811, 4.0 wt% mesogenic cross-linker RM257 (from HCCH), 4.0 wt% 2-ethylhexyl acrylate (from Sigma-Aldrich), and 0.7 wt% photoinitiator Irgacure 651 (from BASF), showing a phase sequence of ISO-(38.1 °C)-BPII-(34.5 °C)-BPI-(27.0 °C)-N*. The precursor was infiltrated into a 12 μm-thick cell and subsequently exposed to ultraviolet light (XLite380, OPAS; centered at $\lambda = 365$ nm) with an intensity of 20 mW cm$^{-2}$ for 60 min to ensure complete polymerization. The PSBP turns from ISO to BPI at ~37.0 °C, and it is well stabilized in the BPI even when the ambient temperature plummets to 13.0 °C (the lower limit of our experimental condition).

**Cells.** Cells are composed of a pair of glass slides with plastic spacers to determine the cell gap: 100 and 300 μm for the polymer-free BPLCs and 12 μm for the polymer-stabilized BPLC demonstrated herein. For samples utilized to perform the experiments shown in Figs. 4–6, the substrates were precoated with polyimide SE-8793 (Nissan) and rubbed with cloth to induce uniform planar alignment.

**Measurements.** Reflection and transmission spectra were taken using a spectrometer USB4000 (Ocean Optics, resolution of ~0.3 nm). Microscopic images/videos and Kössel diagrams were captured using a charge-coupled device DS-Fi1 (Nikon) that was linked to a polarizing optical microscope Eclipse LV100 POL (Nikon) containing a switchable Bertrand lens. In the Kössel diffraction examination, a 440 nm light with a bandwidth of 10 nm was employed as a light source. The Kössel rings were observed in the back focal plane of the objective. Macroscopic images and video were taken using digital single-lens reflex camera D7000 (Nikon) equipped with lens AF-S VR Micro-Nikkor 105 mm f/2.8G IF-ED (Nikon).

**Self-reassembly approach.** The temperature control system implemented was identical to that for the phase sequence measurement (mK2000). For Fig. 1a, **M1** was cooled from ISO with a rate of 10 °C min$^{-1}$ to a specified temperature shown on the horizontal axis, and subsequently sit for 3 h before measurement. For Fig. 1b, c, the holding temperatures were set at 36.6 °C for BPII and 36.0 °C for BPI. Upon quenching from the isotropic phase to a single blue phase or a mixed phase, abundant BP platelets nucleated in the isotropic melt. Before these nuclei grew large, the process was terminated by meeting others (in just a few seconds upon reaching the set temperature). Subsequently, the self-reassembly (platelet-merger) process was initiated. The speed of self-reassembly varies dramatically with the set phase (cf. Fig. 1). The growth process demonstrated in Fig. 2—successive quenching—was accomplished by the following steps. Step 1: Quenching the sample from the ISO to the BPII (36.6 °C) at a rate of 10 °C min$^{-1}$. Step 2: Holding the sample at 36.6 °C (mono-BPII) for 36 h to let the BPII crystals grow spontaneously by platelet merger. Step 3: Using the grown single crystals of BPII as seeds to form large BPI crystals via quenching from the BPII to the BPI (36.2 °C) at a rate of 10 °C min$^{-1}$. Step 4: Holding the sample for another 24 h to let the striations anneal out, releasing the mechanical stress generated during the pseudomorphism process (i.e., Step 3). The data in Fig. 2e, f were collected at 36.0 °C.

**GTS approach.** The experiments were conducted using a temperature-controlled enclosure GS350 (Linkam) and controller T95 (Linkam). For the greenish-colored BPI displayed in Figs. 3–5, **M2** was used, the two temperature-controlled stages were set at 34.5 and 33.0 °C, respectively, and the scan rates were 0.02 μm s$^{-1}$ for Fig. 3 and 0.10 μm s$^{-1}$ for Figs. 4 and 5. For the bluish-colored BPI displayed in Fig. 5, **M3** was used, the stages were set at 32.8 and 32.4 °C, respectively, and the scan rate was 0.10 μm s$^{-1}$. For reddish-colored BPI shown in Fig. 5, **M4** was used, the stages were set at 32.8 and 31.2 °C, respectively, and the scan rate was 0.10 μm s$^{-1}$.

**Data availability.** The data that support the findings of this study are available from the corresponding authors upon reasonable request.

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

## Acknowledgements

This research was funded by Asian Office of Aerospace Research and Development (AOARD), Air Force Office of Scientific Research (AFOSR)—15IOA086; work at NSYSU was partially supported by Ministry of Science and Technology of Taiwan—MOST 104-2628-E-110-003-MY2 and MOST 103-2112-M-110-012-MY3; work at PSU was supported by a grant from AFOSR—FA9550-14-1-0297.

## Author contributions

T.-H.L. conceived the idea. C.-T.H. developed the self-reassembly process, and carried out the experiment with C.-C.L., D.-Y.G., C.-Y.W., and S.-P.J. C.-W.C. and C.-L.H. developed the GTS process. C.-T.H., C.-C.L., C.-W.C., and C.-L.H. carried out the surface-treatment-assisted GTS growth experiment. C.-W.C. and C.-T.H. carried out the polymer-stabilized single crystal experiment. C.T.W. and T.H.L. built the Kössel diffraction system. C.-W.C, H.-C.J., C.-T.H., T.-H.L., and I.-C.K. complete the data analysis. C.-W.C., I.-C.K., T.J.B. and T.-H.L. wrote the paper in collaboration with all the authors. T.H.L., I.-C.K., and T.J.B. co-supervised the project.
