## [Peer Review File · Nature Communications]

Reviewers' comments:

Reviewer #1 (Remarks to the Author):

In this work, the authors fabricated giant single crystals of BPLC. I recommend it for publication in Nature Comm after following questions and comments have been addressed satisfactorily.

- 1) In Fig. 1. Crystal size of BPLC is described. I cannot understand calculating the diameter. How many platelet do you measure? Please explain the equation $D = 2 \times (A/\pi)^{1/2}$.
- 2) How did you measure crystal growth rate?
- 3) In Fig. 5, images of BPI monocrystals with red, green, and blue is depicted. From the images, they seem not to be well-aligned monocrystals because Mura is observed. The image figures should be replaced by well-aligned images without Mura.
- 4) Line 190, you describe that no hysteresis was observed between increasing and decreasing field strength. How did you evaluate the hysteresis? Supporting data should be described?
- 5) Relevant references are not cited in this manuscript. Following papers should be cited?
 - (1) S.-Y. Jo et al., ACS Appl. Mater. Interfaces 2017, 9, 8941–8947
 - (2) S.-T. Hur et al., Adv. Mater. 2013, 25, 3002–3006.
 - (3) K. Kim et al., J. Inf. Disp. 2015, 16, 155–160.
 - (4) K. Kim et al., J. Mater. Chem. C 2015, 3, 5383–5388.

Reviewer #2 (Remarks to the Author):

This paper is an experimental study of the formation and properties of large monodomains of large liquid crystal blue phases. Blue phases have promising applications, particularly in photonics, and the formation of large crystals is important in scaling up these efforts. The work performs a number of interesting experiments demonstrating the processes of assembly into the blue phase II and blue phase I upon quenching from the isotropic phase. Based on observation of self-assembly and reassembly processes in bulk systems, the authors conclude that larger domains may be achieved through growth on a temperature-gradient scanning apparatus that slowly pulls crystals through the phase transition from BPII to BPI, permitting reorganization and the growth of larger crystals. In this way, "world record" size crystals are formed. By tuning the temperature or composition, the feature size of these crystals may be modified to achieve a specific reflectance wavelength. After forming these large domains with liquid crystalline molecules alone, the authors perform photopolymerization to stabilize the BPI phase and modify its properties with an applied electric field. This permits the tuning of reflectance properties for a single crystal formed at given conditions.

The experiments in this paper appear to be carefully performed and the results are interesting to those in the field. It is further not clear what the transformative innovation(s) within this paper are, aside from the achievement of larger crystals than have been achieved before. The work appears to be much more appropriate for a more specialized journal with an emphasis in photonics, or liquid crystals.

1. Taking the first study, described in Figure 1, an isotropic phase is equilibrated, and then quenched to lower temperature, transitioning into either mixed (BPII/Isotropic or BPII/BPI) phases. It is possible to form large BPII domains simply by waiting a significant period of time, while BPI domains do not exhibit a similar property (at least, over a reasonable wait time of less than one week). It is stated then that the use of a large,

monocrystalline BPII domain can aid in formation of larger monocrystalline BPI domains. While some comments are made about the processes which lead to arrest in formation of the BPII phase, no discussion is made about this aside from to show BPII crystals can be enlarged by waiting, and BPI cannot.

- Some important questions about the quenching process for these experiments are mentioned in the Methods section of the paper, but not where the data is presented; if these details are relegated to the methods section, the caption of the figure and the surrounding text should refer to this. Currently, there is a reference to the Methods section at the beginning of the paragraph, but this is inadequate. Such reference is missing in the text surrounding Figure 2 as well.

2. The second piece, described in Figure 2 is a nice demonstration; if one starts with a system that has been rapidly cooled to BPII and then waits for large domains to nucleate, rapid quenching to BPI and subsequent waiting resulted in the formation of larger crystals, which are observed and characterized. It appears from this that the BPI phase may be seeded by the initial formation of the BPII phase. It is not clear from discussion in the paper if this behavior should be unexpected, and thus demonstrates a breakthrough in BPI crystal growth methodology. In Figure 3 involves the use of a temperature gradient method to anneal the formation of BPI crystals and enhance the growth of individual domains.

- If nucleation of BPI from BPII is not as straightforward as stated in the paper, the authors would do well to emphasize the difficulties in obtaining BPI and why other investigations and techniques are insufficient to achieve BPI crystals of similar character.

- The achievement of larger crystals is very useful, and if it were demonstrated to be a robust, scalable method for the production of large crystals, the paper would be significantly more impactful.

3. Armed with the ability to manufacture larger blue-phase crystals, the authors turn their sights to controlling the band gap of the crystals. This point of the paper is a definite strength, and shows that the conditions under which the blue phase is nucleated and the presence of polymer can both be used to engineer specific, sharp band gaps. The studies in this section are very intriguing and perhaps the most impactful results of the paper, since these present the possibility of new photonic technologies; reemphasis of the paper on this result and its implications could, while highly specialized, raise the work to a higher level of impact appropriate for Nature Communications. In its current state, these results seem like an observation tacked on to a paper focused on breaking records for blue phase crystal size.

Ultimately, the issues of clarity may be addressed, but as presented the work does not rise to the level expected for Nature Communications. The bulk of the study is presented as an exercise in patience applied to the nucleation of blue phase crystals, which is more appropriate for a specialized journal. The latter half of the study presents intriguing possibilities for new, transformative technologies, but does not sufficiently discuss them for these results to carry the paper.

Reviewer #3 (Remarks to the Author):

The natural self-assembly of defect-free periodic structures with periodicity in nanometer and micrometer range is a formidable task that may revolutionize the fabrication of photonic crystals. In the manuscript "Large three-dimensional photonic crystals based on monocrystalline liquid crystal blue phases" the authors have developed a gradient temperature method to grow macroscopic monodomains of 3D periodic blue-phase liquid crystals. The presented microscope images clearly demonstrate that monodomains of centimeter size can be grown in the cells of thicknesses of hundreds of microns, stabilized by partial polymerization and tuned by applied electric field. In my opinion, the results revealed by optical images will attract a strong interest of physicists and chemists; however, the reflection spectra, shown in the Figs. 4a, 5b and 6c, do not exhibit the Darwin plateau like the observed ones in the transmission spectra of cholesteric phase, Fig. S2. The absence of the Darwin plateau in the reflection spectra may indicate the presence of distortions, inhomogeneity or defects in periodic structure and be a substantial problem bearing in mind the requirements on periodicity of photonic crystals.

In my opinion, the presented results deserve to be published in Nature Communications, though the absence of the Darwin plateau in the reflection spectra ought to be explained in the revised manuscript.

Sergij Shiyanovskii

The manuscript has been revised and supplemented with new data in reference to the reviewers' comments. Our responses to the comments are summarized for the editor and reviewers as follows.

Reviewer #1

In this work, the authors fabricated giant single crystals of BPLC. I recommend it for publication in Nature Comm after following questions and comments have been addressed satisfactorily.

1) In Fig. 1. Crystal size of BPLC is described. I cannot understand calculating the diameter. How many platelet do you measure? Please explain the equation $D = 2 \times (A/\pi)^{1/2}$.

Response: This is explained in the revised caption of Fig. 1: "The data points for Figs. 1a and 1b were obtained by averaging the areal size of all platelets within the field of view; each platelet is approximated by an enclosing circle of diameter."

2) How did you measure crystal growth rate?

Response: The crystal growth rate is the translational stage scan rate, which is explicitly mentioned in the text (pages 6–7, lines 4–10 of the paragraph right below the caption of Fig. 2).

3) In Fig. 5, images of BPI monocrystals with red, green, and blue is depicted. From the images, they seem not to be well-aligned monocrystals because Mura is observed. The image figures should be replaced by well-aligned images without Mura.

Response: New photos and spectra are inserted in the revised text (cf. Fig. 5). Relevant statements in the main text (page 9, lines 3–5 of the paragraph between Figs. 4 and 5) and Methods section (subsections "Polymer-free BP mixtures" and "GTS approach") have also been revised. Because BPLCs are soft photonic crystals, the Mura caused by slight deformations in a single crystal is hard to avoid in such a thick crystal but hardly degrade the optical performance of the photonic single crystal, such as high reflectivity and sharp bandgap.

4) Line 190, you describe that no hysteresis was observed between increasing and decreasing field strength. How did you evaluate the hysteresis? Supporting data should be described?

Response: Detailed hysteresis measurement and discussion have been presented in Fig. S5 and Section V of the Supplementary Information. The corresponding statement in the main text (page 10, lines 9–10 of the paragraph between Figs. 5 and 6) has also been revised. By repeating the experiments with increasing and decreasing voltages, generally the reflection peak positions remain nearly unchanged. No residual distortion is observed and the hysteresis, defined as the wavelength difference of the PBG at half tuning range between increasing and decreasing field, is only ~ 2 nm. The performance can be further improved by increasing the polymer content and optimize the photo-polymerization conditions.

5) Relevant references are not cited in this manuscript. Following papers should be cited?

(1) S.-Y. Jo et al., *ACS Appl. Mater. Interfaces* 2017, 9, 8941–8947

(2) S.-T. Hur et al., *Adv. Mater.* 2013, 25, 3002–3006.

(3) K. Kim et al., *J. Inf. Disp.* 2015, 16, 155–160.

(4) K. Kim et al., *J. Mater. Chem. C* 2015, 3, 5383–5388.

Response: Reference (2) was quoted in the original manuscript submitted to Nature Communications as Ref. 18. References (1) and (4) are now quoted in the revised introduction section as Refs. 29 and 28.

(1) S.-Y. Jo et al., *ACS Appl. Mater. Interfaces* 2017, 9, 8941–8947

(4) K. Kim et al., *J. Mater. Chem. C* 2015, 3, 5383–5388.

Reviewer #2

2-1. Taking the first study, described in Figure 1, an isotropic phase is equilibrated, and then quenched to lower temperature, transitioning into either mixed (BPII/Isotropic or BPII/BPI) phases. It is possible to form large BPII domains simply by waiting a significant period of time, while BPI domains do not exhibit a similar property (at least, over a reasonable wait time of less than one week). It is stated then that the use of a large, monocrystalline BPII domain can aid in formation of larger monocrystalline BPI domains. While some comments are made about the processes which lead to arrest in formation of the BPII phase, no discussion is made about this aside from to show BPII crystals can be enlarged by waiting, and BPI cannot.

Response: More text has been added to the revised manuscript in the second paragraph of page 4 and the revised caption of Fig. 1. We have also inset two new schematics of the defect structures in unit cells of BPII and BPI, respectively, in Fig. 1b (shown below) to depict the continuous/connected disclinations in BPII and separate disclinations in BPI.

The discussion on why BPII can be enlarged by self-reassembly but BPI cannot is revised and stated in the main text as follows: "*Similar procedures, however, barely yield improvement in the crystalline sizes grown from BPI platelets, similar to other studies employing surface alignment and slow cooling. The difficulty in growing large crystal from the BPI phase compared to BPII is attributable to the fact that within a lattice, all the line defects of BPII intersect with each other thus effectively acting as a single defect, whereas within the BPI phase, the line defects exist independently of one another*^{27,43} (cf. insets of Fig. 1b). The platelet-merger phenomenon follows the system's tendency to minimize free energy. When the lattice contains a sole defect (BPII), only one minimum of free energy exists, at which a single crystal is formed.

In the case of BPI, there exist numerous defects in one lattice which correspond to many local free energy minima."

2-1. (continued) •Some important questions about the quenching process for these experiments are mentioned in the Methods section of the paper, but not where the data is presented; if these details are relegated to the methods section, the *caption of the figure and the surrounding text should refer to this*. Currently, there is a reference to the Methods section at the beginning of the paragraph, but this is inadequate. *Such reference is missing in the text surrounding Figure 2 as well.*

Response: References to the Methods section have been added to line 5 of the third paragraph on page 3 and line 6 of the paragraph on page 5, as well as captions of Figs. 1 and 2. In the subsection "Self-reassembly approach" of Methods, we have added more details on the quenching processes used to obtain the results for Figs. 1 and 2, from lines 4–13:

"Upon quenching from the isotropic phase to a single blue phase or a mixed phase, abundant BP platelets nucleated in the isotropic melt. Before these nuclei grew large, the process was terminated by meeting others (in just a few seconds upon reaching the set temperature). Subsequently, the self-reassembly (platelet-merger) process was initiated. The speed of self-reassembly varies dramatically with the set phase (cf. Fig. 1). The growth process demonstrated in Fig. 2—successive quenching—was accomplished by the following steps. Step 1: Quenching the sample from the ISO to the BPII (36.6°C) at a rate of 10°C/min. Step 2: Holding the sample at 36.6°C (mono-BPII) for 36 hours to let the BPII crystals grow spontaneously by platelet merger. Step 3: Using the grown single crystals of BPII as seeds to form large BPI crystals via quenching from the BPII to the BPI (36.2°C) at a rate of 10°C/min. Step 4: Holding the sample for another

24 hours to let the striations anneal out, releasing the mechanical stress generated during the pseudomorphism process (i.e. Step 3)."

2-2. The second piece, described in Figure 2 is a nice demonstration; if one starts with a system that has been rapidly cooled to BPII and then waits for large domains to nucleate, rapid quenching to BPI and subsequent waiting resulted in the formation of larger crystals, which are observed and characterized. It appears from this that the BPI phase may be seeded by the initial formation of the BPII phase. It is not clear from discussion in the paper if this behavior should be unexpected, and thus demonstrates a breakthrough in BPI crystal growth methodology. In Figure 3 involves the use of a temperature gradient method to anneal the formation of BPI crystals and enhance the growth of individual domains.

- If nucleation of BPI from BPII is not as straightforward as stated in the paper, the authors would do well to emphasize the difficulties in obtaining BPI and why other investigations and techniques are insufficient to achieve BPI crystals of similar character.

Response: The difficulties in obtaining large BPI and why other studies/techniques are unable to achieve large (many-unit-cells) crystals was already explained previously [response to 2-1] and is further elaborated in the revised manuscript (the second paragraph in page 4).

- The achievement of larger crystals is very useful, and if it were demonstrated to be a ***robust, scalable method for the production of large crystals, the paper would be significantly more impactful***

Response: Indeed we have demonstrated that the technique allows polymer-stabilization, leading to the fabrication of more robust crystals with large temperature range and broadband tunability (cf. Fig. 6, the second paragraph of page 10, and Supplementary section V).

2-3. Armed with the ability to manufacture larger blue-phase crystals, the authors turn their sights to controlling the band gap of the crystals. This point of the paper is *a definite strength*, and shows that the conditions under which the blue phase is nucleated and the presence of polymer can both be used to engineer specific, sharp band gaps. The studies in this section are very intriguing and perhaps the most impactful results of the paper, since these present ***the possibility of new photonic technologies; reemphasis of the paper on this result and its implications could, while highly specialized, raise the work to a higher level of impact appropriate for Nature Communications***. In its current state, these results seem like an observation tacked on to a paper focused on breaking records for blue phase crystal size. The latter half of the study presents intriguing possibilities for new, transformative technologies, but does not sufficiently discuss them for these results to carry the paper

Response: We have addressed these general comments in the revised and expanded introduction section (the second paragraph of page 3) and in subsequent relevant results-and-discussion sections, specifically mentioning that the technique developed and the obtained results and insights into self-assembly and reassembly processes to fabricate macroscopic size photonic crystal will impact several other fields as well.

To elaborate their impacts in *optics & photonics*, we add the following statement to the introduction section: "Being able to increase the number of periods in a photonic crystal (N) not only extends the interaction length but also greatly improves the optical properties of a photonic crystal such as the photon density of states at the bandedge (proportional to N^2).³⁵ A significant increase of N from 100 to 10,000 will enhance the density of states by $\sim 10,000$ times, impacting group velocity, lasing threshold, spontaneous emission, and optical nonlinearity.^{4,36,37}" We have also noted from the *material science & topological physics* points of view that, "The colossal, well-oriented BPLCs in either the polymer-free or polymer-stabilized form will serve as excellent templates for more advanced structures and diverse material systems³⁸⁻⁴⁰, as well as topological platforms for molecular self-assembly and symmetry-protected topological photonic crystals^{41,42}"

Reviewer #3:

The natural self-assembly of defect-free periodic structures with periodicity in nanometer and micrometer range is a formidable task that may revolutionize the fabrication of photonic crystals. In the manuscript “Large three-dimensional photonic crystals based on monocrystalline liquid crystal blue phases” the authors have developed a gradient temperature method to grow macroscopic monodomains of 3D periodic blue-phase liquid crystals. The presented microscope images clearly demonstrate that monodomains of centimeter size can be grown in the cells of thicknesses of hundreds of microns, stabilized by partial polymerization and tuned by applied electric field. In my opinion, the results revealed by optical images will attract a strong interest of physicists and chemists; however, the reflection spectra, shown in the Figs. 4a, 5b and 6c, do not exhibit the Darwin plateau like the observed ones in the transmission spectra of cholesteric phase, Fig. S2. The absence of the Darwin plateau in the reflection spectra may indicate the presence of distortions, inhomogeneity or defects in periodic structure and be a substantial problem bearing in mind the requirements on periodicity of photonic crystals.

In my opinion, the presented results deserve to be *published in Nature Communications, though the absence of the Darwin plateau in the reflection spectra ought to be explained in the revised manuscript.*

Response: The sharp reflection peak, rather than a so-called Darwin plateau, is due to the very small index difference in the BP photonic crystal unit cell, unlike cholesteric liquid crystal. We added this explanation to the caption of Fig. 4e.

REVIEWERS' COMMENTS:

Reviewer #1 (Remarks to the Author):

I recommend that this paper is to be published in Nature Comm.

Reviewer #2 (Remarks to the Author):

The authors have done well to address the criticisms in my prior report, as well as those raised by the other two reviewers. Therefore, I recommend this paper for publication.

Reviewer #3 (Remarks to the Author):

In my opinion, the authors have addressed in the revised manuscript the reviewers' questions and suggestions. The manuscript in the current form deserves to be published in Nature Communications.

Sergij Shiyanovskii